# Multi-LexSum: Real-World Summaries of Civil Rights Lawsuits at Multiple Granularities

**Zejiang Shen**[†]    **Kyle Lo**[†]    **Lauren Yu**[◇]    **Nathan Dahlberg**
**Margo Schlanger**[◇]    **Doug Downey**[†,♣]

[†]Allen Institute for AI    [◇]University of Michigan    [♣]Northwestern University
{shannons, kylel, dougd}@allenai.org
{laurenyu, mschlan}@umich.edu nadahlberg@gmail.com

## Abstract

With the advent of large language models, methods for abstractive summarization have made great strides, creating potential for use in applications to aid knowledge workers processing unwieldy document collections. One such setting is the Civil Rights Litigation Clearinghouse (CRLC),[1] which posts information about large-scale civil rights lawsuits, serving lawyers, scholars, and the general public. Today, summarization in the CRLC requires extensive training of lawyers and law students who spend hours per case understanding multiple relevant documents in order to produce high-quality summaries of key events and outcomes. Motivated by this ongoing real-world summarization effort, we introduce Multi-LexSum, a collection of 9,280 expert-authored summaries drawn from ongoing CRLC writing. Multi-LexSum presents a challenging multi-document summarization task given the length of the source documents, often exceeding two hundred pages per case. Furthermore, Multi-LexSum is distinct from other datasets in its multiple target summaries, each at a different granularity (ranging from one-sentence "extreme" summaries to multi-paragraph narrations of over five hundred words). We present extensive analysis demonstrating that despite the high-quality summaries in the training data (adhering to strict content and style guidelines), state-of-the-art summarization models perform poorly on this task. We release Multi-LexSum for further summarization research and to facilitate the development of applications to assist in the CRLC's mission.[2]

## 1 Introduction

Automatic summarization is a longstanding goal of natural language processing. Recently, abstractive summarization methods powered by large pretrained language models have shown impressive results [36, 61]—raising the question of whether these methods can help real-world summarization workloads currently performed by human experts. In this paper, we present a new dataset, Multi-LexSum, for studying automatic summarization in an important real-world application setting found in the Civil Rights Litigation Clearinghouse (CRLC). The CRLC currently collects and presents documents and information from modern large-scale civil rights lawsuits in a manner easily understood by legal practitioners and scholars and the general public alike [10]. Today, the Clearinghouse relies on human legal experts to write summaries of civil rights cases, explaining their events and outcomes. This cognitively demanding task requires summary writers to comprehend multiple documents of different types (often totaling over two hundred pages of text per case); extract entities, events, and their

---

[1]https://clearinghouse.net.
[2]The dataset can be accessed at https://multilexsum.github.io. Please refer to the appendix for details.

36th Conference on Neural Information Processing Systems (NeurIPS 2022) Track on Datasets and Benchmarks.

interrelationships; and synthesize this information into a summary that captures the key details in each case's timeline. For a typical summary, this process takes an expert 1-4 hours. And it needs to be repeated as the case proceeds through the legal system, to keep the summary up-to-date.

Success in summarization automation would allow the Clearinghouse and other efforts like it to greatly increase their coverage and update their summaries in close to real time. Quicker and less costly narrative description of important and routine lawsuits would benefit both the legal field and the general public by increasing access to and understanding of disputes and their resolutions.

We release Multi-LexSum, an abstractive summarization dataset for federal U.S. large-scale civil rights lawsuits drawn from the CRLC. It consists of about 40,000 source documents and 9,000 expert-written summaries (covering about half as many cases). Besides its potential to enable new summarization capabilities to benefit the CRLC effort and others like it, Multi-LexSum has unique characteristics that make it an interesting object of study for summarization research more broadly:

- Unlike some summarization workloads, the CRLC task requires production of summaries at **multiple target levels of granularity**: tiny (25 words, on average), short (130 words), and long (650 words). Variable granularity can be valuable in many applications—e.g., short summaries are ideal to scan for items of interest, and longer summaries to explore more deeply. To our knowledge, Multi-LexSum is the first dataset to provide summaries at multiple levels of granularity. It enables study of multi-task methods that learn from supervision at multiple granularities, and that provide controllable generation at a specified granularity, as our experiments explore.

- Other multi-document summarization datasets offer only 800-8,000 words in the source documents, on average [14, 16, 39], even though many applications require summarizing **large collections of multiple documents**. In Multi-LexSum, the average source length is over 75,000 words.

- Unlike other summarization datasets that are (semi-)automatically curated [11, 22, 26, 28, 35, 44, 52], Multi-LexSum consists of **expert-authored summaries**. The experts—lawyers and law students—are trained to follow carefully crafted guidelines, and their work is reviewed by an additional expert to ensure quality (see Appendix B). This provides high-quality supervision and evaluation and reduces the risk of training on summaries containing facts unsupported by the source text, which can contribute to model hallucination [33, 40].

We conduct a series of experiments on Multi-LexSum, and find that existing summarization models perform poorly. Human assessments of model output result in an average rating of 0.43 on a 0-3 scale, showing that significant improvements are needed before the summaries can provide utility for the CRLC project. Finally, multi-task approaches that train on the multiple granularities of summaries in Multi-LexSum demonstrate promise for improving long summary quality.

## 2 Related work

### 2.1 Natural language processing for legal documents

Much recent work in Natural Language Processing (NLP) has focused on the legal domain [2, 3, 6, 8, 57]. Lawsuits generate rich document sets with domain-specific language and complex structures, which are challenging for state-of-the-art language processing models [3]. Given the important societal role of litigation, along with the extremely high cost of legal expertise, NLP methods to help search, synthesize, and answer questions about legal corpora are of strong interest.

NLP has been applied to a variety of legal document types, including patents [52], legal provisions and contracts [38, 49, 54], legislative bills [34], and court documents [21, 42, 63]. The NLP tasks studied in this work range from document/sentence classification [5, 7, 54] to information extraction [4, 25], question answering [30, 32, 49, 64], and—most relevant to our work—automatic summarization [21, 29, 34, 52]. As found in other specialized domains of language, legal NLP systems often benefit from starting from a large language model pre-trained on legal text [6, 57, 63].

Our Multi-LexSum dataset is focused on automatic summarization of court proceedings and outcomes. Previous work on this task mainly focuses on *extractive* approaches, where the output summaries consist of sentences drawn directly from the source [29]. Hachey and Grover [23] summarize UK court judgments from the HOLJ corpus [21] by selecting the most summary-worthy sentences from a document, while Kim et al. [31] develop a graph-based summary sentence selection method on the same corpus. Yousfi-Monod et al. [60] propose ProdSum, a Naive Bayes sentence classifier, for

Table 1: Three different summaries for one case in Multi-LexSum. We highlight and label spans of text according to which fact it covers.

**Long Summary** L: This case is about an apprenticeship test that had a disparate impact on Black apprenticeship applicants. The [a]Equal Employment Opportunity Commission (EEOC) filed this lawsuit on [d]December 27, 2004, in [e]U.S. District Court for the Southern District of Ohio. Filing [a]on behalf of thirteen Black individuals and [f]similarly situated Black apprenticeship test takers, the EEOC alleged that [g]the individuals' employer, the Ford Motor Company, as well as their union, the United Automobile, Aerospace, and Agricultural implement workers of America (the "UAW"), and the Ford-UAW Joint Apprenticeship Committee, violated [h]Title VII of the Civil Rights Act, 42 U.S.C. §1981, and Michigan state anti-discrimination law. At issue were the selection tests for apprenticeship training programs, whose disparate impact denied Black applicants eligibility and admission. [i]The EEOC sought injunctive relief, as well as damages (including backpay) for the Black apprenticeship applicants. The case was assigned to [j]Judge Susan J. Dlott.

[k]The individuals also brought a separate class action against Ford and the UAW, Robinson v. Ford Motor Company, (No. 1:04-cv-00844), and the cases were consolidated on January 6, 2005. As a result, the case was transferred to [j]Judge S. Arthur Spiegel. Six months later, in June 2005, both cases were resolved via a [l]class settlement agreement. Ford agreed to pay $8.55 million and to implement a new selection process for its apprenticeship programs. This agreement further required Ford to hire an industrial psychologist to design this new selection process and to place 279 members of the settlement class on the eligibility list for the Ford apprenticeship program.

On [m]June 15, 2005, the court found that the proposed settlement agreement was fair. [n]2005 WL 5253339. The next day, the court ordered that Ford pay $1.1 million to cover attorneys' fees and expenses incurred during settlement negotiations, and $567,000 to cover fees and expenses associated with the implementation and monitoring of the settlement agreement. [n]2005 U.S. Dist. LEXIS 12071. As [o]the settlement was initially scheduled to last for three years, and [p]there is no further activity on the docket sheet, this case presumably closed in 2008.

**Short Summary** S: This case is about an apprenticeship test that had a disparate impact on Black apprenticeship applicants. The Equal Employment Opportunity Commission (EEOC) filed this lawsuit on December 27, 2004, in U.S. District Court for the Southern District of Ohio. Filing on behalf of thirteen Black individuals and similarly situated Black apprenticeship test takers, the EEOC alleged that the individuals' employer, the Ford Motor Company, as well as their union, the United Automobile, Aerospace, and Agricultural implement workers of America (the "UAW"), and the Ford-UAW Joint Apprenticeship Committee, violated Title VII of the Civil Rights Act, 42 U.S.C. § 1981, and Michigan state anti-discrimination law. The EEOC sought injunctive relief and damages for the Black apprenticeship applicants. The individuals also brought a separate class action against Ford and the UAW, and the cases were consolidated. In June 2005, both cases were resolved via a class settlement agreement. Ford agreed to pay $8.55 million and to implement a new selection process for its apprenticeship programs, and the court ordered Ford to cover attorneys' fees and expenses. This case is closed.

**Tiny Summary** T: 2005 class action settlement resulted in Ford paying $8.55m to redesign its selection process for apprenticeship programs to address the previous process's disparate impact on Black applicants.

| **Checklist of Facts during Writing** | | | |
|---|---|---|---|
| a. Plaintiff description | e. Court's full name | i. Remedy sought | m. Date of settlement/decree |
| b. Type of counsel | f. Class description | j. Judge's name | n. Citation to an opinion |
| c. Type of action | g. Defendant description | k. Consolidated case | o. How long decrees lasted |
| d. Filing Date | h. Statutory basis for case | l. If class action | p. Last action in case |

summarizing case decisions from the Canadian Legal Information Institute (CanLII). Galgani et al. [18] experiment with citation based summarization approaches on case reports from the Australasian Legal Information Institute AustLII [20]. Systems like CaseSummarizer [47], LetSum [17], and the pipeline by Zhong et al. [65] are developed to extractively summarize documents from AustLII, CanLII, and Board of Veterans' Appeals (BVA), respectively. By contrast, our work focuses on *abstractive* summarization, where the target output is a paraphrase of the source, creating the potential for providing more succinct summaries in more accessible language. We release a large dataset of over 9,000 expert-written summaries for court documents from about 4,500 U.S. federal civil rights lawsuits, as a testbed for state-of-the-art abstractive summarizers.

## 2.2 Summarization datasets in other domains

Multi-LexSum contains expert-written summaries of up to three different granularities for the same source; this is the first such published dataset to the best of our knowledge. Perhaps the most similar

work is BookSum [35]; however in contrast to our work, BookSum's multiple summaries consider different lengths of the source to be summarized—paragraphs, chapters, and the whole content in a book. Multi-LexSum presents a new opportunity to study how to learn from and produce summaries at varying granularity for the same source, as we explore in our experiments.

Another key differentiating factor in Multi-LexSum is that its summaries are expert-provided. In order to scale to impressive sizes, many existing summarization datasets are created in a (semi-)automatic fashion—e.g., using the first sentence [44] or summary bullets [43] as the target summary for a piece of news, or automatically extracting and linking scientific paper abstracts [11] and citing sentences [14]. These datasets lack a clear specification of how the summary corresponds to the source, can have varying quality, and often contain information that is not directly supported or implied by the source, which can degrade the factual consistency of models trained on the data [33, 40]. By contrast, Multi-LexSum contains "gold" summaries. Experts are specifically trained to write the case summaries following carefully crafted instructions (detailed in Appendix B), and the written summaries are subsequently reviewed to ensure correctness and stylistic consistency.

Compared to many existing single- or multi-doc summarization datasets for news [15, 43, 44], scientific papers [11, 14, 39], patents [52], legislative bills [34], and government reports [28], the summary context in Multi-LexSum comes from multiple sources that are extraordinarily long—over 75k words, an order of magnitude larger than most other datasets (see Table 2). One exception is BookSum [35], which uses entire books as summary inputs; the books are on average 127,000 words long. However, it has far fewer samples (403) than Multi-LexSum does (4,500).

## 3 Multi-LexSum

### 3.1 Task definition

In the American legal system, civil lawsuits ("cases") involve a set of actions among two or more parties and the judge(s) [12]. Most steps in the case are taken by way of formal document filings. The first step typically occurs when the "plaintiffs"—people, groups of people, or entities—file a "complaint" against one or more "defendants" in a state or federal trial court. The case then proceeds as the parties file additional documents. It is through these documents that the parties lay out the case background, explain their arguments, rebut opposing parties' arguments, and ask for specific actions from the judge(s) (see Table 7 for a breakdown of document types). The judge(s) also file documents which set schedules, ask questions, and memorialize rulings—intermediate orders that frame the conflict or instruct parties to take various steps or "final" orders that at least temporarily resolve the case [13]. All a result, a case's documents can extend to hundreds, even thousands, of pages of text. Collectively, these documents paint a full picture of the case, but they can be extremely time-consuming to read and digest in order to understand the gist.

The goal of legal case summarization is to write a short article that captures principal details and describes each case's litigation history in plain language—information that is otherwise often difficult to come by. The CRLC summaries come in three different lengths:

- **Long** (L) summaries typically contain multiple paragraphs, covering the case background, parties involved, and proceedings. Major case events and outcomes typically receive a paragraph each.
- **Short** (S) summaries have only one paragraph with a shorter description of the background, parties involved, and the outcome (so far) of the case.
- **Tiny** (T) summaries are one-sentence summaries intended to appear on Twitter to describe the case at a specific point in its history.

Using the different granularities of summary, we define a variety of distinct summarization tasks. First, we consider three different multi-document summarization (MDS) tasks that map from the source documents D to each of the summary lengths above (e.g., D → L denotes the task of mapping the source documents to the long summary). We also consider three different single-document summarization (SDS) tasks that take a ground truth summary as input and attempt to map to a shorter summary as output (e.g., S → T denotes mapping from a short summary to the corresponding tiny one). Finally, the multiple granularities in Multi-LexSum create the opportunity to use *sets* of the data as input or output, which we also explore (e.g. {L, D} → T denotes taking a long summary and the source documents as input, and outputting a tiny summary). Sometimes (part of) the input text may be generated from another model, which we denote using a prime symbol, e.g., {L′, D} → T.

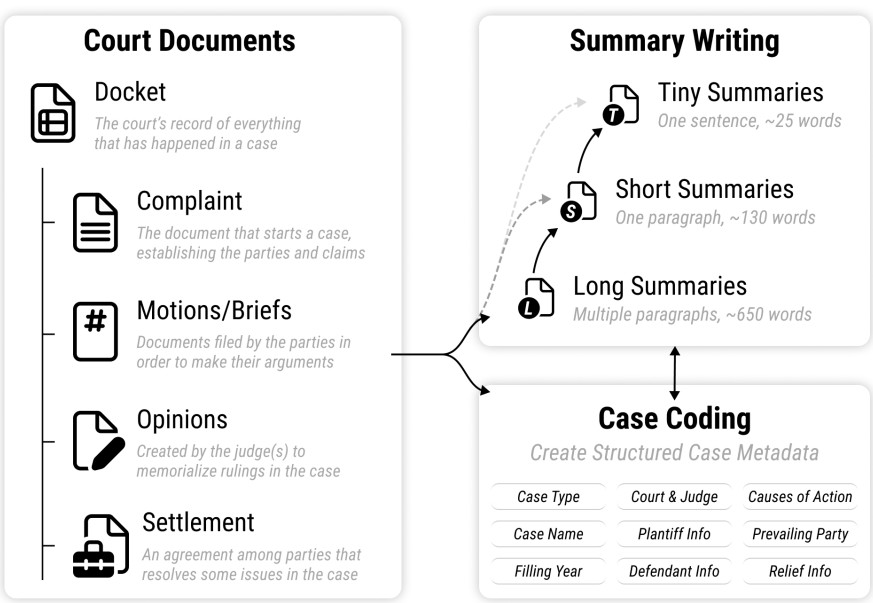

Figure 1: The pipeline of creating summaries and structured data for lawsuits.

## 3.2 Creating Multi-LexSum summaries

All the data in Multi-LexSum, including the selected documents, summaries, and the structured case metadata, are manually curated by legal experts: legal scholars, attorneys, and law students who receive specialized training relevant to their CRLC assignments. For a case where not much has happened since the lawsuit was filed, it typically takes one to two hours for an inexperienced law student to read source documents and write the summary. Summarization of more developed cases requires more time—around two to four hours. Even an experienced attorney might spend ten or more hours to understand and summarize an unusually complex case.

Figure 1 illustrates the summary writing pipeline. After receiving a specific lawsuit assignment, the summary writer reads through court documents, especially the docket, which contains a chronological list of every document filed (Appendix B.1). From the massive document collection, the summary writer selects a small subset of documents (on average, eight) that provide information about major events, and attaches them to the case in CRLC.

The summary writing then takes place, guided by the instructions defined in Appendix B.2. To ensure the coverage of the principal information in a case, writers can resort to a checklist of facts they need to include for the case. They typically write the long summary first, and create the short and tiny versions after, with the option to refer to the source documents as well as the longer summaries. Cases can last a long time—sometimes several decades—so the summaries may be updated with new material several or many times as the case progresses. Table 1 shows summaries for the case *EEOC v. Ford Motor Company*.

After a summary writer has completed a draft of a summary, another lawyer or law student with additional experience and specialized training reviews the summary for accuracy and readability. When needed, the reviewer edits the summary to ensure it is factually correct and conforms with the writing style guideline (Appendix B.3).

## 3.3 Dataset characterization

Table 2 compares key measurements between Multi-LexSum and other SDS and MDS datasets. We report dataset sizes and the average number of source documents per sample (which is 1 for SDS datasets). To calculate average number of words and sentences in the source document(s) and target summaries, we use the SpaCy library [27] `en_core_web_sm` model. Finally, we provide average extractive fragment coverage, density, and compression ratio, as defined by Grusky et al. [22]. Multi-LexSum is distinct in that its source text and long target summaries are much lengthier

Table 2: Comparison of Multi-LexSum to other single-document (SDS) and multi-document (MDS) summarization datasets. Measurements include dataset size, number of source documents per sample, number of words and sentences in source and target texts, and source-target coverage, density, and compression ratio. Except for number of samples, all reported values are averages across all samples, including test sets when available.

| Dataset | Samples | Source | | | Target | | Source →Target | | |
| --- | --- | --- | --- | --- | --- | --- | --- | --- | --- |
| | | Docs | Words | Sents | Words | Sents | Coverage | Density | Compress |
| **Tiny** | | | | | | | | | |
| **XSUM** | 226,677 | 1 | 454.5 | 31.7 | 24.0 | 1.0 | 0.67 | 1.10 | 19.9 |
| **SciTLDR** | 3,229 | 1 | 5847.7 | 232.3 | 22.2 | 1.1 | 0.95 | 4.85 | 310.8 |
| **Newsroom** | 1,212,739 | 1 | 800.1 | 37.9 | 31.2 | 1.5 | 0.83 | 9.53 | 43.6 |
| **BookSum/Paragraph**[1] | 147,665 | 1 | 163.131 | 8.411 | 35.8 | 1.9 | 0.51 | 0.90 | 6.7 |
| **Multi-LexSum** D → T | 1,603 | 10.7 | 119072.6 | 5962.5 | 24.7 | 1.4 | 0.92 | 2.27 | 5449.6 |
| **Short** | | | | | | | | | |
| **BigPatent** | 1,341,362 | 1 | 3629.0 | 131.4 | 116.7 | 3.5 | 0.86 | 2.38 | 36.8 |
| **MS^2** | 16,212 | 24.0 | 7775.6 | 306.3 | 65.1 | 3.9 | 0.86 | 1.91 | 174.8 |
| **Multi-XScience** | 40,528 | 5.1 | 817.0 | 32.0 | 119.7 | 4.9 | 0.67 | 1.30 | 7.7 |
| **CNN / Daily Mail** | 311,971 | 1 | 805.2 | 39.3 | 59.9 | 6.1 | 0.85 | 3.49 | 14.9 |
| **BillSum** | 23,455 | 1 | 1804.1 | 54.4 | 218.4 | 6.4 | 0.90 | 4.05 | 12.9 |
| **Multi-LexSum** D → S | 3,138 | 10.3 | 99378.2 | 5017.0 | 130.2 | 5.1 | 0.96 | 3.33 | 840.7 |
| **Long** | | | | | | | | | |
| **Multi-News** | 56,216 | 2.8 | 2168.1 | 92.2 | 264.0 | 10.4 | 0.83 | 5.01 | 8.2 |
| **BookSum/Chapter** | 12,570 | 1 | 5339.6 | 302.1 | 421.0 | 21.7 | 0.78 | 1.47 | 16.6 |
| **BookSum/Book** | 403 | 1 | 126537.2 | 6964.2 | 1163.1 | 56.0 | 0.90 | 1.79 | 146.3 |
| **Multi-LexSum** D → L | 4,534 | 8.8 | 75543.2 | 3814.2 | 646.5 | 28.8 | 0.94 | 4.07 | 97.4 |

[1] The BookSum number might be slightly different from those reported in original paper because some samples weren't successfully downloaded using the script provided by the authors.

than existing datasets, with the exception of BookSum[3] which has far fewer samples and focuses on the literature domain. Long source text poses a challenge for identifying the salient information to include in the summary.

We find that Multi-LexSum's summaries have a high fraction of terms that also appear in the source, but are still abstractive. We follow Grusky et al. [22]'s approach that analyzes the coverage and density based on extractive fragments, which are shared spans of tokens that can be jointly identified in the document and summary. Multi-LexSum has the top coverage for long summaries, of 0.94, meaning that 94% of the words in the summary can be found in the extractive fragments from the corresponding source documents. The generally high coverage for all Multi-LexSum granularities suggests that its summaries contain fewer unsupported entities and facts compared to the datasets with lower coverage.[4] At the same time, the density (or average length of the extractive fragments) ranges from 2-4 for Multi-LexSum, suggesting that most of the summary sentences are not verbatim extractions from the sources and are instead abstractive.

## 4  Experiments

Our experiments on Multi-LexSum focus on two questions: (1) can models generate and synthesize information from the massive source documents in MDS tasks (D → L, D → S, and D → T); and (2) can models be configured to produce summaries of the desired lengths and details for SDS tasks (L → S, L → T, and S → T)?

---

[3]We compare with the full book summarization task in BookSum given it has similar source/target lengths.

[4]Though it is worth mentioning that as long as the ground truth summaries do not include unsupported entities or terms, we would expect the coverage to become high as the length of the source documents increases, because there's a higher chance that words in the summary will happen to appear in the source documents. High coverage scores does not imply that the task is easy: models still need to find which words to use and how to compose them in the summary.

Table 3: Performance of baseline models on different MDS tasks in Multi-LexSum.

| Models | D → L | | | | | D → S | | | | | D → T | | | | |
|---|---|---|---|---|---|---|---|---|---|---|---|---|---|---|---|
| | $R\text{-}1_{f1}$ | $R\text{-}2_{f1}$ | $R\text{-}L_{f1}$ | $BS_{f1}$ | Words | $R\text{-}1_{f1}$ | $R\text{-}2_{f1}$ | $R\text{-}L_{f1}$ | $BS_{f1}$ | Words | $R\text{-}1_{f1}$ | $R\text{-}2_{f1}$ | $R\text{-}L_{f1}$ | $BS_{f1}$ | Words |
| First $k$ Sent | 30.41 | 9.67 | 14.50 | 9.73 | 813.7 | 21.97 | 7.17 | 13.61 | -1.60 | 120.5 | 8.69 | 1.07 | 6.81 | -8.72 | 33.9 |
| Random $k$ Sent | 35.62 | 9.15 | 13.78 | 10.91 | 838.7 | 24.37 | 3.79 | 12.92 | 5.28 | 141.4 | 10.39 | 0.84 | 7.78 | 0.91 | 45.9 |
| BERT-EXT | 36.94 | 10.65 | 14.43 | 12.85 | 860.3 | 24.79 | 5.48 | 13.89 | 4.49 | 137.3 | 9.74 | 1.12 | 7.50 | -5.43 | 39.3 |
| PEGASUS | 40.79 | 20.01 | 25.36 | 34.83 | 203.8 | 43.35 | 19.91 | 29.99 | 37.88 | 94.6 | 22.61 | 7.09 | 18.44 | 26.78 | 22.3 |
| BART | 48.79 | 23.78 | 28.73 | 39.55 | 351.3 | 43.55 | 19.98 | 29.84 | 37.41 | 114.0 | 23.28 | 8.19 | 19.36 | 27.84 | 21.7 |
| LED-4096 | 47.75 | 24.13 | 28.89 | 39.10 | 295.0 | 45.44 | 21.00 | 30.99 | 39.33 | 103.1 | 26.52 | 8.92 | 21.47 | 30.01 | 22.4 |
| LED-16384 | 49.07 | 25.17 | 29.40 | 40.05 | 310.1 | 46.54 | 22.08 | 31.91 | 40.00 | 108.0 | 28.11 | 9.84 | 22.88 | 31.41 | 21.5 |
| PRIMERA | 53.73 | 27.32 | 30.89 | 42.01 | 416.3 | 45.51 | 21.04 | 30.81 | 39.32 | 110.2 | 27.56 | 9.26 | 21.84 | 30.40 | 27.9 |

## 4.1 Experimental Setup

We split all cases into train (70%, 3177 samples), Dev (10%, 454), and Test (20%, 908). All cases have long summaries, and 70% and 36% of the cases have short or tiny summaries, respectively. The corresponding source and target document lengths are reported in Table 2. Appendix E provides extra details about split sizes and how the splits are determined.

**Models** We experiment with summarization models that are representative of the state-of-the-art. BART [36] and PEGASUS [61] are two recent abstractive summarizers based on the Transformer architecture [55] and have achieved state-of-the-art performance on multiple summarization datasets. Owing to the large multi-document source content in Multi-LexSum, we also experiment with two recent summarizers tailored to this setting. Longformer Encoder and Decoder (LED) [1] and PRIMERA [58] are two models that can handle longer inputs (16384 and 4096 tokens, respectively) by introducing sparsity into attention layers, and PRIMERA adds an MDS-specific pre-training objective to improve performance on MDS tasks.

**Implementation and Computational Resources** For abstractive summarizers, we finetune the models based on the PyTorch [46] implementations from the HuggingFace library [56]. For each task, the models are trained for 6 epochs on two RTX A6000 GPUs from an internal cluster, with a learning rate of 5e-5.[5] Following previous work [35], we use beam search with 5 beams and n-gram repetition blocks for n>3 when decoding the generation outputs. The total GPU hours used for training all the benchmark models sum to roughly 300 hours. Additional training details are available in Appendix F.

**Automatic Evaluation** ROUGE-{1,2,L} [37] and BERT Score (BS) [62] are used to compute the lexical and estimated semantic overlap between the generated and gold summaries. We use the DeBERTA [24] model for sentence embedding following the authors' suggestion. We report the average of F1 measures for ROUGE and BS on the test set, and include the number of generated words for reference.

## 4.2 Multi-doc legal case summarization

Table 3 lists model performances on the three MDS tasks, in which the summarizers are challenged to fetch key information from the extraordinarily long input strings. We test a set of extractive baselines: following [51], we develop two extractive heuristics that select the first $k$ or random $k$ sentences from the source documents ($k = 35, 6, 2$ for L, S, T, respectively). We compare them with the BERT-Extractive-Summarizer (BERT-EXT) [41], which embeds the source documents using sentence Transformers [50] and selects $k$ key sentences from the embedding clusters. The best performing extractive models are worse than the abstractive counterparts (PEGASUS) by 47%, 64%, 84% in terms of ROUGE-2, and the magnitude increases as the target is more abstractive. Because the sentence embedder is not trained for legal text, BERT-EXT attains similar (or worse in the case of D → S) performance to the two extractive heuristics.

---

[5]For LED training, we use 3e-5 following authors' recommendations [1] and gradient checkpointing [9] to reduce GPU memory usage.

Table 4: Model performance for generating shorter summaries from the longer version. Comparing with Table 3, performance is much higher when the model is given a ground truth summary of a different size as input.

(a) Fine-tuning PEGASUS and BART on SDS tasks.

| Models | R-1$_{f1}$ | R-2$_{f1}$ | R-L$_{f1}$ | BS$_{f1}$ | Words |
|---|---|---|---|---|---|
| L → S Fine-tuning | | | | | |
| **PEGASUS** | 54.32 | 35.62 | 42.58 | 47.49 | 156.8 |
| **BART** | 56.04 | 37.02 | 44.16 | 49.19 | 133.8 |
| L → T Fine-tuning | | | | | |
| **PEGASUS** | 32.86 | 14.44 | 27.20 | 34.62 | 24.6 |
| **BART** | 31.65 | 13.05 | 25.52 | 33.59 | 24.0 |
| S → T Fine-tuning | | | | | |
| **PEGASUS** | 34.15 | 16.15 | 28.27 | 34.73 | 25.6 |
| **BART** | 34.02 | 15.20 | 27.96 | 35.48 | 24.4 |

(b) PRIMERA models results on progressive summarization.

| Target | R-1$_{f1}$ | R-2$_{f1}$ | R-L$_{f1}$ | BS$_{f1}$ | Words |
|---|---|---|---|---|---|
| {L, D} → S | | | | | |
| **Gold** L | 54.99 | 36.42 | 43.44 | 48.69 | 133.4 |
| **Predicted** L′ | 41.41 | 18.24 | 27.53 | 34.04 | 164.0 |
| {L, D} → T | | | | | |
| **Gold** L | 34.07 | 14.84 | 27.74 | 36.13 | 24.13 |
| **Predicted** L′ | 23.63 | 7.98 | 19.50 | 27.09 | 24.05 |
| {S, D} → T | | | | | |
| **Gold** L | 34.60 | 16.50 | 28.71 | 35.62 | 28.65 |
| **Predicted** L′ | 22.50 | 6.79 | 18.01 | 25.88 | 27.86 |

For abstractive summarizers, models that allow long inputs (LED and PRIMERA) perform better than BART and PEGASUS (with only 1024 input tokens at most) on all three tasks, indicating the helpfulness of the longer input context. Because LED and PRIMERA models provide pre-trained weights with different max input lengths (16384 and 4096, respectively), we test two variants of LED (LED-16384 and LED-4096) with the corresponding input lengths. The longer input length brings consistent performance improvements for LED across the three tasks, ranging from +4% to +10% of ROUGE-2 in D → T. PRIMERA outperforms even the LED-16384 model on the D → L task, but achieves similar results as LED-4096 on the other two tasks of shorter targets, aligned with the authors' observation [58].

All the summarizers fail to generate long summaries of lengths that match the human summaries—PRIMERA produces the longest summaries of 416 words on average, less than 65% of the ground-truths' average length of 647—while their generations for short and tiny summaries can match the gold label lengths (130 and 25 words on average). This highlights the limitations of existing summarizers in producing long abstractive summaries, as required for Multi-LexSum.

### 4.3 Generating shorter summaries from the longer version

To further explore the multiple granularities of summary in Multi-LexSum, we train summarizers to generate shorter summaries from the longer versions. Shown in Table 4a, models trained on the L → S, L → T, and S → T task show significant improvements compared to their D → ∗ counterparts: for example, the ROUGE-2 of PEGASUS is improved by 79%, 104%, and 128%, respectively, and exceeds scores from "long-input" models like LED and PRIMERA. The model performance in S → T is better than L → T, providing further evidence that inputs with more condensed information simplify the summarization task.

The high summary quality when condensing long summaries to shorter ones suggests a strategy for leveraging training summaries at multiple granularities—a pipeline approach where one model generates a long summary, which is used as input in another model to generate a shorter summary. As an initial test, we train PRIMERA models for each of {L, D} → S and {S, D} → T, which generate a short/tiny summary based on the corresponding long/short summary and the source documents. We use ground-truth summaries L and S for training. Illustrated in Table 4b, when also provided with the gold long/short summaries in the input at test time, PRIMER matches the performance of the counterparts reported in 4a. However, when we use the model in a pipeline that does not assume a ground truth summary as input, substituting it with a BART-generated one, the performance degrades and can be worse than the corresponding D → ∗ models by more than 20% (when generating tiny summaries T).

### 4.4 Multitask training for summaries of different lengths

Another strategy for leveraging summaries at multiple granularities is to train *one* model that can create summaries of different lengths. We indicate the desired summary using prefixes [48], prepending

Table 5: Comparing BART performance under multitask and single-task scenarios. The three-task model improves performance over single-task models.

| Lengths | Samples | R-1$_{f1}$ | R-2$_{f1}$ | R-L$_{f1}$ | BS$_{f1}$ | Words | Samples | R-1$_{f1}$ | R-2$_{f1}$ | R-L$_{f1}$ | BS$_{f1}$ | Words |
|---|---|---|---|---|---|---|---|---|---|---|---|---|
| | **BART, Multitask: D → {L, S, T}** | | | | | | **BART, Single-task: D → L, D → S, D → T** | | | | | |
| L | 6517 | 47.89 | 23.24 | 28.31 | 39.16 | 336.6 | 3177 | 40.79 | 20.01 | 25.36 | 34.83 | 203.8 |
| S | 6517 | 43.80 | 20.14 | 29.89 | 38.00 | 122.6 | 2210 | 43.35 | 19.91 | 29.99 | 37.88 | 94.6 |
| T | 6517 | 25.38 | 8.92 | 20.91 | 29.11 | 23.1 | 1130 | 22.61 | 7.09 | 18.44 | 26.78 | 22.3 |
| | **BART, Multitask: L → {S, T}** | | | | | | **BART, Single-task: L → S, L → T** | | | | | |
| S | 3340 | 55.20 | 36.11 | 43.42 | 48.53 | 133.5 | 2210 | 56.04 | 37.02 | 44.16 | 49.19 | 133.8 |
| T | 3340 | 32.51 | 13.68 | 26.46 | 35.22 | 23.1 | 1130 | 31.65 | 13.05 | 25.52 | 33.59 | 24.0 |

one of "summary: long", "summary: short", or "summary: tiny" to the input source when for generating L, S, or T summaries, respectively. Table 5 compares this multitask model with its single task counterparts. We fix the same number of training epochs, thus multitask models are trained for more steps; however, increasing the number of steps was not found to improve the single-task models.

We find training for three rather than two different tasks generally leads to better performance. The added training samples bring greater performance boosts for T summarization, which has only a third of the training samples compared to L. S summary results are not improved much over single-task when using three tasks, and are slightly worse using two tasks. Most interestingly, L summarization is greatly improved (by 11-17% in the automated metrics) in the three-task case. Since all training cases in the dataset have a long summary, the only difference in the multi-task training is that the model is exposed to the short and tiny views of the summary for the same cases.

# 5 Human evaluation

To assess the usability of these models, we conduct an evaluative study with law students trained to contribute summaries to the CRLC. In all, despite iterative efforts to improve system performance, we found today's models struggle to perform the task well.

**Study Design** In coordination with the CRLC, we developed the following study setting. (1) We scoped to only the D → L setting, which is the most effort-intensive and could benefit the most from model-backed assistance. (2) We used a BART model to generate summaries. (3) Participants included two CLRC writers who edited the generations to produce summary text for 40 new cases that aren't present in Multi-LexSum; this process took them 180 hours in total. (4) We recorded edits made to the summaries as well as asked writers to rate the generation on a 4-point scale.[6]

**System Design** Initial feedback from CRLC experts indicated that the end-to-end generated output of long summaries were too far from usable. Notably, they tended to hallucinate key information (e.g., filing date or court's name), and the experts stated it would take longer to correct errors than to write the summary from scratch. So we designed an alternative system based on iterative CRLC expert feedback. System features included (1) a tool for writers to select relevant text snippets while reading source documents, to aid the model in salient information selection and (2) model generation of each summary paragraph separately based on selected snippets. Given that this system was developed in conjunction with CRLC stakeholders and greatly simplified the computer task to improve performance, we view it as a more accurate reflection of how modern summarization methods might be used in real-world applications. It thus serves as a reasonable tool to assess the usability of these models. Further details about this system can be found in the Appendix C.

**Results** Comparing the generations to post-edited summary texts, ROUGE-1, ROUGE-2, ROUGE-L and BERT-Scores were 45.6, 30.0, 35.4 and 38.0, respectively; these scores are similar to those of BART from our D → L experiments presented in 3. Yet, the system generations received a 0.43 user rating, demonstrating the significant limitations of automated performance metrics. Writers averaged 87 token edits per paragraph, 76% the average length of paragraphs, and they on average extend

---

[6]The rating levels were 0 (bad; completely unusable); 1 (somewhat helpful but requiring >50% edits); 2 (requiring <50% edits); 3 (perfect; no edits needed).

generation lengths by 65%. Follow-up interviews indicated the problem of erroneous or missing key fields continued to prevent the generations from being useful.

## 6 Conclusion

In this paper, we introduce Multi-LexSum, an abstractive summarization dataset for large-scale civil rights lawsuits from U.S federal courts. Multi-LexSum is packed with unique features, including summaries of multiple levels of granularity for the same source, large collections of long source documents, and expert-authored summaries. Through a series of experiments, we find existing summarization models struggle to produce the summaries directly from the long source documents. The average rating of 0.43 on a 0-3 scale from human assessments of current models also suggests substantial room for improvement.

Multi-LexSum is not without its limitations. CRLC is more likely to include cases where the plaintiff wins because such cases typically last longer and receive more attention. This project is further limited to federal cases for which dockets are available online. Performance might not generalize to under-represented cases (e.g., where the defendant wins); we additionally provide case metadata to facilitate future diagnosis of this bias.

We hope Multi-LexSum will aid development of real-world summarization systems intended to assist the activities of both specialized projects like the CRLC as well as more general sites geared toward dissemination of court documents for the general public, e.g., `https://www.courtlistener.com/recap/`. More broadly available and up-to-date case descriptions would be of enormous assistance to reporters, advocates, and members of the general public. The benefit would be even greater for larger "free law" projects that post information about hundreds of thousands, rather than thousands, of lawsuits.

## Acknowledgements

We thank the reviewers for their very helpful suggestions and feedback! We thank the following institutions and entities who generously provide the support for the curation of the underlying Civil Rights Litigation Clearinghouse data over its 15-year history, including: University of Michigan Law School; Washington University in St. Louis School of Law; Center for Empirical Research in Law; Arnold Ventures, "Improving Criminal Justice Reformers' Use of Litigation Information, Documents, and Insights" (2021-2023); Vital Projects Fund, "Revamping the Civil Rights Litigation Clearinghouse" (2021); Proteus Fund, "Revamping the Civil Rights Litigation Clearinghouse" (2021); National Science Foundation SES-0718831, "The Litigation Process in Government-Initiated Employment Discrimination Suits" (2007). The construction of the Multi-LexSum dataset was also funded in part by NSF Convergence Accelerator Award ITE-2132318.

We thank the hundreds of law student authors of the case summaries in Multi-LexSum, listed in `https://clearinghouse.net/people`. We also appreciate the advice from Adam Pah, Arman Cohan, Iz Beltagy, John Giorgi, Lucy Lu Wang, Jonathan Bragg, Dan Weld, Sida Li, and Ruochen Zhang.

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
