# OpenReview forum: "Multi-LexSum: Real-world Summaries of Civil Rights Lawsuits at Multiple Granularities"
_NeurIPS.cc/2022/Track/Datasets_and_Benchmarks — NeurIPS 2022 Datasets and Benchmarks _

### Official Review · Reviewer_VoTk · 2022-07-07
**Good dataset on a policy-relevant problem that fills multiple gaps**

**Rating:** 7
**Confidence:** 4
**Clarity:** Yes, the paper is well written and pa…

**Strengths:**

The dataset is to my knowledge a unique collection of expert-written summaries of very long documents. The summaries are presented at varying levels of granularity. The data addresses an important social problem. The experiments demonstrate the value of the data. The authors demonstrate a real-world application with feedback from human subjects.

**Weaknesses:**

I am uncertain about the data quality of the source documents. I was not confident that the experiments displayed the best possible versions of the models.

**Additional Feedback:**

Congrats on really nice work!

**Correctness:**

With the sole exception of the data quality question for the source documents, everything else seemed to be in great shape. The code repo looks good. I think putting the data on HF would be good.

**Documentation:**

Yes, the authors do a nice job of describing how the data is organized and the documentation is good. Also they have an extensive dataset decription in the appendix that they can translate easily into a model card on HF.

**Ethics:**

No.

**Relation To Prior Work:**

Yes.

**Summary And Contributions:**

Thanks for the opportunity to review this paper. I recommend accepting.

The authors introduce a new set of summaries of long legal documents concerning civil rights cases. To my knowledge, a set of expert-written summaries of this magnitude is unique for a legal application, especially because many of the previous projects cited rely on data that is subject to extremely restrictive licensing regimes (CanLII and AusLII). And this data is focused on a subject area with high potential for social impact. The existence of multiple levels of granularity is a very nice feature. This is not only useful for legal applications,; many people are increasingly aware of the need to handle long sequences, and this will be a nice task to push models in that direction. The authors' point on the length of output sequences from SOTA models speaks to this point; this data will be really useful in future efforts to solve that problem. Also, I was especially impressed by the HCI contribution, which is all too rare in studies like this one. The paper is clearly written.

Some issues to be aware of. My #1 question is about data quality. In the example notebook, the snippet of the source document suggests serious parsing errors ("DIscTcquR FOR THE MIDDLE DISTRICTLO ALAAMA"). These are not atypical for OCR projects but I wonder what the prevalence of errors like these is. **Please help clarify this in your responses.** Pervasive parsing issues would make learning the summaries difficult and reduce the value of the data. As for the benchmark results, I wasn't sure why the authors chose to finetune for only 6 epochs or the choices that were made to handle other specific elements (e.g. unique legal vocab, citations, etc.) that might have impeded performance. More detail on the HCI summarization project would have been nice to include in the main paper as that's a unique and important element. Overall, I think these problems can probably be solved.

---

> ### Author Response · Authors · 2022-08-15
> **We thank you for the constructive feedback and we address your major concerns.**
>
> We thank you for the constructive feedback. In response to your major concerns:
>
> **The quality of source document text**
> > (review quote) In the example notebook, the snippet of the source document suggests serious parsing errors ("DIscTcquR FOR THE MIDDLE DISTRICTLO ALAAMA"). These are not atypical for OCR projects but I wonder what the prevalence of errors like these is.
>
> - Only 11.3% of the source documents in our dataset are scanned documents, with the rest being born-digital PDFs. Given that, we believe the overall quality of the source text is high.  We will release a tag for each source document indicating whether it is scanned or not.
> - To enable further improvements, we will also release the source PDFs (as indicated in the dataset sheet). Researchers can use different OCR methods / PDF parsers on those documents to potentially improve results. Additionally, we hope that supplying the PDF documents can support research on multi-modal document understanding and summarization, starting from raw PDFs.
>
> **Additional experiments for the baseline models?**
> > (review quote) As for the benchmark results, I wasn't sure why the authors chose to finetune for only 6 epochs or the choices that were made to handle other specific elements (e.g. unique legal vocab, citations, etc.) that might have impeded performance
>
> - In our revisions, we will expand discussion of how we made these choices.  We do not perform any special processing for unique aspects of our corpus (such as citations or legal terminology), but these are interesting possibilities for future improvements.  For our baseline models and settings, we choose models that are state-of-the-art in similar summarization dataset papers (e.g. BookSum, Multi-XScience, GovReport) and additionally add models like PRIMERA and LED that can handle long input context which is a feature of our dataset.  For our training settings, we searched different hyperparameter configurations on BART and found the model validation performance converged after 6 epochs for training. We then used the same set of training hyperparameters on all the experiments for both BART and PEGASUS. The hyperparameters for "long models" like LED and PRIMERA are slightly different, based on recommendations in the respective papers. We release our model training scripts for reproducibility.

---

### Official Review · Reviewer_8TcV · 2022-07-15
**A good multi-doc summarization dataset based on civil rights lawsuits**

**Rating:** 7
**Confidence:** 3
**Correctness:** The dataset is constructed in a sound…

**Strengths:**

1.	From the perspective of document summarization, Multi-LexSum firstly provides summaries at multiple levels of granularity.
2.	From the perspective of legal AI, the proposed Multi-LexSum consists of legal expert-authored summaries, and would encourage legal AI studies, especially the applications such as legal case summarization etc.
3.	The experiments were conducted by considering both the automatic evaluations and the human evaluations.

**Weaknesses:**

1.	Some details on data pre-processing, especially the data selection, are not introduced clearly. The dataset is drawn from the ongoing CRLC writing. What features are used for selecting/filtering the data records from the whole posts? Are they selected based on the charge, year, or other features? Sometimes, the inappropriate data selection criteria may result in a dataset with potential social biases.

2.	Multi-LexSum is based on large-scale civil rights lawsuits. It depends on the lawyers and law students to produce high-quality summaries of key events and outcomes. Is there any critical difference between the general-purpose long document summarization and the summarization in Multi-LexSum? Are there any legal-domain specific designs in Multi-LexSum due to the special properties of lawsuits?

3.	The details on the annotation are not provided. For example, how many annotators are spent on one document? How about the inter-annotator agreement and how to measure the agreement?


**Additional Feedback:**

There are some typos. For example, in Line-197, “Test(30%, 908)” --> “Test(20%, 908)"

**Clarity:**

the paper is well-organized and easy to follow.



**Documentation:**

The paper provides a link to the data repository.

**Ethics:**

I see no ethics issue.

**Relation To Prior Work:**

The paper discussed natural language processing for legal documents and summarization datasets in other domains.

**Summary And Contributions:**

This paper introduces Multi-LexSum, a multi-doc summarization dataset based on civil rights lawsuits from U.S. federal courts.

Compared to existing benchmarks, Multi-LexSum provides three granularities (tiny, short, and long) of summaries for source documents. It enables the studies of multi-target summaries at these three granularities. The paper also conducted a series of experiments on Multi-LexSum. The results indicate that exiting popular summarization models cannot achieve satisfying performances.

---

> ### Author Response · Authors · 2022-08-15
> **We appreciate your constructive feedback and we clarify your questions in the response!**
>
> **More details about dataset construction**
>
> > (review quote) The dataset is drawn from the ongoing CRLC writing. What features are used for selecting/filtering the data records from the whole posts? Are they selected based on the charge, year, or other features? Sometimes, the inappropriate data selection criteria may result in a dataset with potential social biases.
>
> - We do recognize the importance of detailed dataset documentation, especially rubrics for selecting samples in the dataset. We detail how the CRLC cases are sampled in the paper: Section 6 (Line 312) in the main paper and the composition section in the dataset sheet (under the title of "Does the dataset contain all possible instances or is it a sample (not necessarily random) of instances from a larger set?" on page 11 of the [supplementary material](https://openreview.net/attachment?id=z1d8fUiS8Cr&name=supplementary_material)).
>
> - The cases in Multi-LexSum are a subset of those in CRLC: to ensure the high quality of the source document text, we have two selection criteria: only federal cases are included, and only those that have a docket that is at least partially available in the online federal court docket system, “PACER”. Generally, federal cases that were ongoing after 2003 meet these criteria. No other special filtering was applied.
>
>
>
> **Unique properties of law in Multi-LexSum compared to general purpose summarization datasets?**
>
> > (review quote) Is there any critical difference between the general-purpose long document summarization and the summarization in Multi-LexSum? Are there any legal-domain specific designs in Multi-LexSum due to the special properties of lawsuits?
>
> - Yes, there are critical differences between Multi-LexSum and other "general-purpose" long document summarization datasets. Compared to many existing resources that are "artificially" curated, i.e., generated by automatic methods or designed as proxy tasks by NLP researchers rather than by domain experts, our dataset is rooted in a real-world task, with carefully designed writing guidelines and high-quality expert-created samples. Moreover, as most of the available summarization datasets (e.g., BookSum, GovReport, WikiSum, Arxiv, CNN/Daily Mail) are focused on general domains like news, literature, wiki, or science, Multi-LexSum is unique in its focus on a particular niche, and presents an opportunity to evaluate how well "general-purpose" models perform in this area.
>
> - We also observe unique structure in the legal documents/text that is not present in existing datasets. For example, court documents fall into distinct types that serve different purposes and should be read differently (see table 7 in the supplementary materials). The “docket” is the most prominent example; it includes a timeline for all the major events and usually serves as a starting point for human writers understanding the case. Other documents contain certain details or supporting facts that should be extracted and used in the summary. Studying specialized models that leverage these domain-specific document structures is an item of future work.
>
> (See additional responses in the next comment)

---

> > ### Author Response · Authors · 2022-08-15
> > **Additional Response**
> >
> > **Detailed information about the annotation process**
> >
> > > (review quote) The details on the annotation are not provided. For example, how many annotators are spent on one document? How about the inter-annotator agreement and how to measure the agreement?
> >
> > - We pay particular attention to documenting the dataset curation process; due to space constraints, they are included in Appendix B from page 16 to page 20 in the [supplementary material](https://openreview.net/attachment?id=z1d8fUiS8Cr&name=supplementary_material)). You can find an overview of the annotation pipeline and the detailed description of each step.
> >
> > - In our dataset, multiple writers (or annotators) can contribute to or update the same summary as the case progresses in court, but it is not necessary for CRLC to have multiple people write different versions of the same summary.  As we discuss in the submission, each summary is written by one expert and reviewed by another for quality, providing some assurance of a level of annotator agreement on the final summaries in the dataset.  Quantifying the level of agreement is difficult.  More generally, in the field of document summarization, how to determine annotator agreement is still an unanswered question, and none of the previous work listed in table 2 reports any inter annotator agreement measures about the target summaries.
> >
> > - We want to highlight that the legal document summarization is **a real-world task**: the design and annotation collection process is thus different from a typical data collection pipeline. Typical data collection processes rely on hiring annotators to work on new tasks that are designed for a research purpose. Therefore, it is not uncommon to have multiple annotators to work on the same instance as an attempt to gauge and reduce the human errors. However, in our case, the annotation task is a real-world process—all of the summary writers are trained or in-training law professionals, the CRLC has developed a systematic work pipeline for the summarization effort, and the summaries are posted online and read by other law practitioners. As such, we consider each summary to be a "gold" annotation: the training and style guidelines ensure consistency, while the reviewing process aims to ensure the factuality and correctness of the summaries.

---

### Official Review · Reviewer_Aeqk · 2022-07-25
**Multi-LexSum dataset releases 4500 U.S. federal civil rights lawsuits and 9000 summaries**

**Rating:** 5
**Confidence:** 5
**Clarity:** Yes, the paper is well organized and …

**Strengths:**

 1 – The dataset creates new challenges for the current text summarization models.
2 – The paper conducted a good series of experiments to support their claim.
3 – The dataset can enhance the summarization models for the law and civil rights domain


**Weaknesses:**

Although the paper has conducted a series of good experiments, but it misses many crucial points that weakens the paper’s claim.  In the paper, there are many ambiguities and unclear statements that hinder the delivery of data construction, experiment settings and data evaluation.
For data collection and construction, in page two the authors mention that there are about “40,000 source documents and 9,000 expert-written summaries”. Then, in page 4 the number of the samples are less than 4600. There is a great confusion here. Besides, it is not clear that whether each document has multiple summaries or multiple documents that have one or more summary. For example, in table 2 the number of samples for tiny summaries for documents/multi-documents is 1603, 3138 for short summaries, and 4534  for long summaries. The average documents for those summaries vary between 8-10 documents. It is better to show that how many documents that only have one tiny, short, or long summary or one document can have many summaries.  Is each summary derived from more than one document? How many summaries that are only derived from one single document? In addition, there is a misleading claim regarding the size of the dataset. The paper compares the law-specific domain summaries (which is acceptable) to Newsroom and other large text summarization datasets. Newsroom, CNN/Daily Mail and Xsum are far larger in size. How can you claim that your data is large compared to those datasets?
There are other ambiguities in the data evaluation regarding the characteristics of the summaries. The paper claims that their summaries are abstractive because they are written by authors which is undeniable. However, when looking at the Coverage scores of their analysis to the characteristics of their data the numbers are concerning. The coverage score quantifies how much the summary borrows words from the original document. Multi-Lex Sum average scores range between 0.92-0.96 which indicates a very high number of words borrowed from the document. It is true the sentences are not extracted directly from the document but there are not new words that could represent the summary. Almost all the words are present in the original document which makes them extractive but not necessarily organized in the same order. The Density measure could support the author’s claim because the numbers are not very high. It is better to include figures to show the scores of coverages and density.
The other concerning point is the human evaluation of dataset. The first attempt did not succeed in page 8, so the authors designed an alternative system to score their human evaluation, but the system is not clearly described, and the results are briefly described which casts some doubts and unclarity on the human evaluation method. What is the specific objective of the human evaluation? Is there a small example of the evaluation in the paper? The detailed examples can be added in the appendix, but the main explanation and short examples should be present in the paper. The intended measurements of Human evaluation are missing.
For the experiment setting: The paper tries to generate more succinct version (tiny or short summaries) of the long summaries and the documents.  What is the objective of this experiment? Multi-LexSum has already multiple granularities. The experiment of generating summaries from summaries casts doubts on the evaluation. Are the summaries generated only from the original document, from the original document + long summary, or only from the long summary? Each scenario has its own follow-up questions for evaluation.


**Additional Feedback:**

No

**Correctness:**

Yes, the dataset is built in a correct and sound method. The evaluation methods can to be revised and the claim of abstractive summaries can be reviewed.

**Documentation:**

Yes

**Ethics:**

Yes

**Relation To Prior Work:**

Yes, the paper mentions most available works, but it is confusing that sometimes the paper compares to BookSum as the closest work to theirs but the paper does not mention how much it is different from it. The conducted comparison to other relevant works are the same as BookSum even though BookSum is the most closely related to their work. More comparison is expected from the most similar works.

**Summary And Contributions:**

Multi-LexSum is a law-specific domain multi-document text summarization dataset. The summaries are claimed to be abstractive. The summaries in this dataset vary from long, short, and tiny and derived from multiple documents. The paper conducts series of experiments to show how challenging the dataset is on the current state-of-the-art text summarization models.
The contribution of the paper is as follows:
1 – The summaries have multiple-granularities which means that there are long, short and tiny summaries.
2 – The summaries are written by law experts.
3 – The dataset includes a fair number of multiple documents and their summaries

---

> ### Author Response · Authors · 2022-08-15
> **Thank you for the constructive questions regarding our paper. We clarify your questions below and will improve our paper to minimize possible future confusion accordingly**
>
> Thank you for the constructive questions regarding our paper -- we will improve our paper to minimize possible future confusion accordingly. We provide detailed comments to your question as follows:
>
> **Clarification of sample sizes in the dataset**
>
> > (review quote) For data collection and construction, in page two the authors mention that there are about “40,000 source documents and 9,000 expert-written summaries”. Then, in page 4 the number of the samples are less than 4600. There is a great confusion here. Besides, it is not clear that whether each document has multiple summaries or multiple documents that have one or more summary, …
>
> - Our task is multi-document summarization, which means that (1) for each case, there is more than one source document and (2) the goal is to write summaries for the case based on all the source documents (instead of writing individual summaries for each document). Therefore, there are more source documents (40,119) than the number of cases (4539).
> - For each case, there are one or more summaries of different lengths. Each case has a long summary, and can optionally have short or tiny summaries, or both of them. The average number of summaries we have per case is 2. Therefore, there are more summaries (9280) than the number of cases (4539). A detailed breakdown of summary lengths is available in table 9 in the supplementary materials.
>
> **Clarification of how Multi-lexsum is "large"**
>
> > (review quote)  In addition, there is a misleading claim regarding the size of the dataset. The paper compares the law-specific domain summaries (which is acceptable) to Newsroom and other large text summarization datasets. Newsroom, CNN/Daily Mail and Xsum are far larger in size. How can you claim that your data is large compared to those datasets?
>
> - We detail in Table 2 the relative size of our data set compared to other summarization data sets.  We would not claim that our data set is larger than those you mention, but there are two ways in which it is, in our view, “large.”:
>     - It is large for a collection of gold, i.e. manually written and curated, summaries.  The massive summarization datasets (Newsroom, CNN/Daily Mail and Xsum) by contrast are automatically constructed (by e.g., extracting the first sentence of a piece of news), which can lead to inconsistent quality, as we note.
>     - The source document sets are large, as quantified in Table 2.  The only dataset with a comparable number of source tokens per summary is BookSum, and that has an order of magnitude fewer samples than our dataset.
>
> **Clarification of the interpretation of the coverage score in Table 2**
>
> > (review quote)  The paper claims that their summaries are abstractive because they are written by authors which is undeniable. However, when looking at the Coverage scores of their analysis to the characteristics of their data the numbers are concerning. ....
>
> - We agree that the density scores help establish that the summaries are abstractive (see final paragraph of Sec 3).  You raise an important point about interpreting the coverage metric, which we will clarify in our revisions.  As long as the ground truth summaries don’t include unsupported entities or terms, we would expect the coverage to become high as the length of the source documents increases, because there's a higher chance that words in the summary will happen to appear in the source documents. As Table 2 in the paper suggests, it's typical to observe high coverage scores in datasets with long input like BookSum (for book-level, Cov=0.90, source words=126537.2 ), BigPatent (Cov=0.86, source words=3629.0). The correlation between the number of words in the source document and the coverage score is r = 0.49.
>
> **Clarification of the human evaluation conducted in our work**
>
> > (review quote) The other concerning point is the human evaluation of dataset... What is the specific objective of the human evaluation? Is there a small example of the evaluation in the paper? ...
>
> - The objective of the human evaluation is to measure to what extent existing summarization models can be helpful for the CRLC’s summarization effort in practice.  We will state this more explicitly in the paper.
> - Given the scope of the paper, we choose to include a very detailed description of the human evaluation system in the Appendix C from page 20 to 22 in the [supplementary material](https://openreview.net/attachment?id=z1d8fUiS8Cr&name=supplementary_material).
> - We learned from the domain experts that an end-to-end method that directly generates summaries from all the source documents is not helpful for the evaluation (as they expect rewriting the whole model outputs is often needed), therefore we collaborated with the domain experts to design a simplified version of summary task and evaluate the model performance, which is detailed in Appendix C.
>
> (See additional responses in the next comment)

---

> > ### Author Response · Authors · 2022-08-15
> > **Additional response**
> >
> > **Clarification of the objectives for the "summarizing summaries" experiments**
> >
> > > (review quote) For the experiment setting: The paper tries to generate more succinct version (tiny or short summaries) of the long summaries and the documents. What is the objective of this experiment? Multi-LexSum has already multiple granularities. The experiment of generating summaries from summaries casts doubts on the evaluation. Are the summaries generated only from the original document, from the original document + long summary, or only from the long summary? Each scenario has its own follow-up questions for evaluation.
> >
> > - The objective of the "summarizing summaries" is multi-fold:
> >     - From a real-world perspective, summaries of different granularities are needed by the lawyers: the lawyer will choose to read summaries of different lengths under different scenarios. Right now all the summaries have to be manually written by law students and legal professionals, and even having a model to produce a shorter summary from the longer version could save them a lot of writing time.
> >     - From a research perspective, it is interesting to explore whether having some middle-level supervision (e.g., the long summary) can improve the generation of a more succinct summary (e.g., the short summary).  In table 4, we contrast different scenarios, i.e., having a model to summarize a "gold" longer version (table 4(a)), summarize a gold long + source document (table 4(b)) and summarize a generated long + source document (table 4(b)). We find that the intermediate supervision is very helpful and yields better results.
> >     - In the paper, we use small cap letters to denote the "gold" summaries (e.g., L, S, T), and add a prime symbol to mean generated summaries from the source (e.g., L', S'). We will clarify this notation in the revision to make it clearer.

---

### Official Review · Reviewer_en1S · 2022-07-26
**A well-crafted resource with excellent documentation**

**Rating:** 9
**Confidence:** 4
**Correctness:** The dataset construction and experime…

**Strengths:**

* Since labels are difficult to acquire in law, datasets with expert annotations (such as this one) are immensely valuable.
* The task of generating a summary from multiple lengthy documents is mainly untouched but extremely important. As the authors point out, such an automatic summarization would help legal experts in drafting lawsuit descriptions more efficiently, which would benefit citizens in accessing and understanding more legal disputes and resolutions.
* The authors perform an extensive analysis of existing summarizers on their dataset.
* The paper is a joy to read, very clear and well structured.
* The dataset documentation is thorough.

**Weaknesses:**

The extent of novelty, technical or scientific depth is limited, yet, appropriate for a resource paper.

**Additional Feedback:**

All in all, the paper is very well written, and the dataset is unique and valuable.

**Clarity:**

The paper is very well written and structured. Also, it is easy to follow, understand, and good to read.

**Documentation:**

The dataset is publicly available under the Open Data Commons Attribution License (ODC-By) license. The authors provide a hosting and maintenance plan in Appendix A.2, detail the data annotation guidelines in Appendix B, and give a thorough dataset datasheet in Appendix F.


**Ethics:**

Some ethical concerns are mentioned in Appendix D. No other salient concerns come to mind.

**Relation To Prior Work:**

The related works are accurately described in the second section.

**Summary And Contributions:**

The authors present a new dataset of 9,280 expert-authored summaries from large-scale civil rights lawsuits that include three types of summary (tiny, short, and long). They experiment with state-of-the-art models for abstractive summarization (BART, Pegasus, LED, and PRIMERA) on their new dataset and show that these models perform poorly on lengthy summaries.

---

> ### Author Response · Authors · 2022-08-15
> **Thank you for your review!**
>
> Thank you for your review! We are glad that you enjoyed reading our paper and acknowledged the possible impact our work can bring to the community. We will continue working on the dataset in the future.

---

### Official Review · Reviewer_V3B5 · 2022-07-28
**Great dataset**

**Rating:** 8
**Confidence:** 4
**Correctness:** Yes, all seems good.

**Strengths:**

__Significance of contributions__

I think the paper has two strengths.

First, from the perspective of the legal NLP community, the dataset that the authors introduce is incredibly valuable for several reasons:

1. Legal NLP benchmarks are *extremely* difficult to build, given how expensive legal annotations are. This is a very high quality dataset, on US case law (also rare!). An open question in legal NLP is whether pretraining on legal texts actually helps performance. A large difficulty in answering this question is the lack of datasets corresponding to “hard” tasks. This seems like a task/dataset which could really push our understanding forward.
2. Summarizing legal texts is a non-trivial task which is a core task that many lawyers do. Law students are required to summarize cases every day in class. Judges summarize cases in legal opinions. Lawyers summarize cases when constructing arguments in court and in legal briefs. Thus, this task could enable tools which have a meaningful impact.

From the perspective of the machine learning/NLP community, the task that the authors identify–multi-granularity summarization–seems understudied (based on my skim online). Thus, this datasets would be of value to the broader NLP community (and not just those focused on legal applications).


__Rigor__
The experiments are well done and reasonable in light of the claims made by the paper.

**Weaknesses:**

I don’t really see any major weaknesses!

**Additional Feedback:**

One thing I’m curious about–do the authors have a sense of which documents are most important for producing a summary? My intuition tells me that complaints and opinions would be most valuable, in part because they already summarize the important aspects of a case (e.g. the facts section, the procedural posture section, etc).

**Clarity:**

Yes, the paper is well written. I have a background in law though, so I’m not sure if some of the legal jargon would be difficult for a non-law CS person to understand.

**Documentation:**

Yes.

**Ethics:**

This is all good.

**Relation To Prior Work:**

I really like the positioning between (1) prior work on legal NLP, and (2) summarization datasets. This is very well done.

**Summary And Contributions:**

The paper first introduces and describes Multi-LexSum: a benchmark for multi-document summarization and three levels of granularity (L, S, T). Multi-LexSum consists of the documents comprising the docket for cases. The task is to summarize the facts and procedural history of the case. The paper then evaluates a range of models on Multi-LexSum. They explore summary-summary summarization (i.e. going from a long summary to a short summary) and multi-task summarization. Finally, they measure human-assess quality of the summaries using domain experts, and discuss how such tools can be incorporated into real world pipelines.

---

> ### Author Response · Authors · 2022-08-15
> **Thank you for your review! And we find indeed source documents contribute to the summary differently based on their types.**
>
> Thank you for your review! We appreciate your constructive feedback, and it's great to learn that you enjoyed reading our paper. Regarding your question:
>
> **Source documents contribute to the summary differently based on their types**
> > (review quote) One thing I’m curious about–do the authors have a sense of which documents are most important for producing a summary? My intuition tells me that complaints and opinions would be most valuable, in part because they already summarize the important aspects of a case (e.g. the facts section, the procedural posture section, etc).
>
> - Your intuition is correct.  We conducted the following experiments to verify your thoughts, where we choose to remove a specific type of document from the summary input and train and evaluate the models ([**see the result table here**](https://multilexsum.github.io/assets/remove-doc-exp-table.png)). Firstly removing any document from the source input will negatively impact the output summary quality. Comparing removing the complaint, docket, opinion, or a random document (of other types), it seems complaint and opinion have a stronger influence on the summary quality. We see that the rouge score after removing a complaint or an opinion is significantly (p<0.10) lower than the full-document rouge, whereas it is not significantly lowered by removing a random other document or even (perhaps surprisingly) the docket.
> - When we remove multiple document types from the input, there's also a more significant drop of rouge scores when complaint and opinion are removed together. Interestingly, despite the test model being a LED-16384 with longer input context, the summary quality is worse than the LED-4096 model when either complaint or opinion is not present.
> - Taken together, the experiments demonstrate the unique challenges in this dataset for processing legal documents, and suggest the potential of domain-specific modeling techniques.

---

> > ### Comment · Reviewer_V3B5 · 2022-08-16
> > **Response**
> >
> > That's really interesting! Thanks for sharing.
> >
> > Congratulations again on a nice piece of work. I'm looking forward to seeing what others can do with this dataset.

---

### Author Response · Authors · 2022-09-02
**Updates to the paper**

We would like to thank all the reviewers for their constructive feedback. We address the concerns in the comments for each reviewer individually, and we’ve updated the paper and supplementary materials following their suggestions. The updates include:

1. Add additional experiment results that ablate the importance of the source documents of different types [Reviewer 1(V3B5)]
2. Clarification on the interpretation of the coverage scores in footnote 4 [Reviewer 3(Aeqk)]
3. Add one illustrative example of an end-to-end generated long summary with expert annotations on spans of texts with potential errors. in table 8 in the Appendix. [Reviewer 3(Aeqk)]
4. Add description in Appendix F for how the document texts are extracted and preprocessed from the source PDFs; especially we detail the portion of the scanned document and how we obtain the text. [Reviewer 5(VoTk)]
5. Include more details about the choices of hyper-parameters and used models in Appendix F. [Reviewer 5(VoTk)]
6. Clarification and typo fixes
    - Fix the typo mentioned by Reviewer 4(8TcV)
    - Clarify what the “prime” notation means in line 149 [Reviewer 3(Aeqk)]
7. Other changes
    - Include the exemplar short and tiny summary in the Table 1
    - Move the figure of “The pipeline of creating summaries and structured data for lawsuits.” to the main paper (Figure 1).

---

### Meta-Review · Program_Chairs · 2022-09-16

**Recommendation:** Accept
**Confidence:** 4

**Metareview:**

This is a very high-quality dataset, which is especially noteworthy since legal NLP benchmarks are very difficult to build. The authors approached every aspect of the process with extreme care. It will likely have an impact on law practice and start interesting discussions about the use of ML in these settings. The paper is also very well written and enjoyable to read.

Most reviewers are heavily in favor of acceptance. Reviewer Aeqk brought up some weaknesses, but at least from the AC's perspective, these seem to be answered well by the authors.

---

### Decision · Program_Chairs · 2022-09-16

Accept